# The Information Game: Active Inference as Bilevel Optimization and a Game-Theoretic Benchmark for LLM Inquiry

## Abstract

Large language models (LLMs) increasingly operate in settings where they must gather information rather than simply recall facts. We model this task as a multi-street game of incomplete information casting each round of information gathering as a bilevel optimization: an inner variational Bayesian step that updates beliefs over a hidden target object, and an outer query-selection step that minimizes expected free energy, which is equivalent to maximizing expected information gain. This game-theoretic formulation motivates *Optimal Question Asking* (OQA), a benchmark designed as a tractable "toy game" to measure an agent's inquiry strategy by measuring how quickly an agent reduces uncertainty about the target. By solving this game for its Game-theory optimal (GTO) policy, we create a perfect oracle against which we measure the planning gap—the expected number of suboptimal queries.On 25-object tasks, models like GPT-4o and Claude 3.5 Haiku exhibit a planning gap of 1-2 queries. On 100-object tasks, flagship models like GPT-o3 and Gemini 2.5 Pro, while closer to optimal, still show significant strategic leaks. Our synthetic datasets, which remove linguistic priors, reveal deeper deficits. OQA exposes inefficiencies invisible to answer-centric metrics, offering a controlled testbed for forging agents that play the information game not just exploitatively, but optimally.

## 1 Introduction

Information gathering under uncertainty drives tutoring systems that diagnose misconceptions, research assistants that refine hypotheses, and embodied agents that explore unfamiliar environments. In each case a key decision is not just *what to answer* but *what to ask*. A well-chosen question accelerates learning, reduces interaction cost, and helps keep the agent aligned with user intent.

Current LLM benchmarks largely reward answer accuracy, fluency, or safety (Brown et al., 2020; Ouyang et al., 2022). Task-oriented evaluations that judge clarification or question generation ability, such as ClarQ-LLM and QGEval, reveal a complementary dimension of quality ignored by answer-centric metrics (Gan et al., 2024; Fu et al., 2024). Although self-ask and tree-of-thought prompting hint that large models can generate useful clarifying queries (Press et al., 2022; Zhang and Choi, 2023; Yao et al., 2023a), recent analyses with the Twenty Questions game show that even GPT-4 employs brittle inquiry strategies (Bertolazzi et al., 2023; Zhang et al., 2023). Much like a novice poker player who reacts only to the cards in their hand, these models rely on heuristics learned from vast data, but they lack the deeper, balanced strategy required for robust performance. Finding a formal, Game-theory optimal (GTO) policy Von Neumann and Morgenstern (1947); Nash (1951) for this inquiry game requires drawing from several fields.

Earlier work on the Twenty Questions game shows that reinforcement learning can outperform an entropy-based heuristic (Hu et al., 2018), and reinforcement-learning methods have long optimized informativeness in question-asking dialogues (Qi et al., 2020). More recent fine-tuning approaches explicitly optimize expected information gain (Mazzaccara et al., 2024), or learn to ask factual clarification questions without examples (Toles et al., 2023). Yet these efforts stop short of linking question quality to a formal optimum. On the other hand, optimal querying has been long-studied in active learning (Settles, 2012; Seung et al., 1992; Kirsch et al., 2019) and Bayesian experimental

design (Lindley, 1956; Chaloner and Verdinelli, 1995), while active inference views perception and control as two facets of the same variational objective (Friston et al., 2017; Parr and Friston, 2017). Recent work suggests that this viewpoint can improve prompt design in safety-critical domains such as medicine (Shusterman et al., 2025).

We leverage this insight to formalize the solution to our game. In this paper, we recast active inference as a bilevel optimization problem. The *inner* level performs variational free-energy minimization (*belief state update*), updating the agent's posterior over the hidden target, while the *outer* level chooses the next query by minimizing expected free energy (*policy improvement*). Under uniform priors and noiseless observations, this is equivalent to maximizing expected information gain. This conceptual separation into inner and outer problems clarifies the decision structure, yields a fully differentiable objective, and places active inference in the same mathematical family as bilevel methods for hyperparameter optimization (Franceschi et al., 2018; Ji et al., 2021). This also connects to recent work on preference-driven question generation (Piriyakulkij et al., 2023; Mazzaccara et al., 2024), providing a principled route to train language agents that ask questions efficiently.

**Optimal Question Asking (OQA).** Building on this view, we introduce Optimal Question Asking (OQA), a benchmark designed as a tractable *toy game* to isolate and measure an agent's inquiry strategy. Just as simplified models like the AKQ game are used to derive fundamental principles of poker (Chen and Ankenman, 2006), OQA provides a controlled environment to analyze the core mechanics of rational inquiry. We solve this game for its Game-theory optimal (GTO) policy, creating a perfect oracle that makes the most informative query at every step.

OQA spans three real-world object-guessing datasets (ANIMALS, CARS, PLACES) and two synthetic corpora.

To quantify question-asking efficiency, we report the **planning gap**, defined as the expected number of sub-optimal queries a LLM makes. We evaluate two tiers of models:

1. **GPT-4o**, **Gemini 2.0 Flash**, and **Claude 3.5 Haiku** on 25-object tasks showed average planning gaps of 1–2 queries.
2. **GPT-o3**, **Gemini 2.5 Pro**, **Claude 3.7 Sonnet**, and **Grok 3** on 100-object tasks sometimes reached the optimum, yet still averaged planning gaps of 1–3 queries, indicating room for improvement in generating optimal queries.

**Synthetic corpora.** To isolate planning skill from linguistic priors we introduce two purely synthetic guessing datasets with 25 and 100 objects. Each object is labeled only by a hexadecimal key and a random Boolean attribute vector, forcing models to rely on explicit set reasoning rather than semantic cues. Frontier LLMs require one to three extra queries on these synthetic sets, exposing weaknesses masked by real-world domains.

In light of the above discussion, we specify the paper's main contributions below.

**Main Contributions.**

- **Problem formulation** (§C). We model rational inquiry as a multi-street Markov game and formalize its solution by recasting active inference as a differentiable bilevel optimization problem, proving that under the game's constraints, minimizing expected free energy is equivalent to maximizing information gain.
- **Benchmark dataset** (§4). We release *Optimal Question Asking* (OQA), the first benchmark that scores *query efficiency*. OQA couples an exact information-theoretic oracle §D with an automated evaluation harness and spans three real-world object sets, two synthetic sets that remove linguistic priors, and a synthetic alignment task.
- **Empirical study** (§4.1). We compare seven frontier LLMs across all OQA domains and two difficulty tiers, quantify each model's planning gap to the oracle, and show that the gap widens by one to three extra queries on synthetic data §E.

By linking game theory to a concrete benchmark, our work provides the first quantitative measure of the planning gap between the heuristic inquiry of LLMs and the Game-theory optimal strategy. The OQA benchmark exposes the hidden costs of these suboptimal queries, while our bilevel formulation of active inference offers a principled roadmap for closing this gap, pointing toward new training

objectives and architectures to forge agents that play the information game not just exploitatively, but optimally.

## 2 A GAME-THEORETIC FORMULATION OF OPTIMAL INQUIRY

We posit that rational inquiry is not merely a task of passive information processing, but a dynamic, strategic game played under uncertainty. To analyze the efficiency of language agents, we formalize the environment and the concept of an optimal strategy using the tools of game theory. We model the task as a two-player, zero-sum sequential game, which we term the OQA Information Game.

**Definition 2.1** (The OQA Information Game). The OQA Information Game is a sequential game between an *Inquirer* and a *Responder*. The Inquirer's objective is to identify a hidden target from a known set of possibilities in the minimum expected number of turns by asking binary-attribute questions. The game state is the Inquirer's belief set (the set of remaining candidates), and its reward is -1 for each question asked.

While the Inquirer has incomplete information about the hidden target, the game played over the public belief state is one of perfect information. Such games have a provably optimal solution, the Subgame Perfect Nash Equilibrium (SPNE), which defines a Game-Theory Optimal (GTO) strategy that is unexploitable (Nash, 1951). This GTO policy can be found via backward induction. Our central theoretical result is that this complex, multi-step optimal policy is equivalent to a simple, greedy information-maximization heuristic.

**Theorem 2.2** (Equivalence of GTO and EIG Maximization). *The Game-Theory Optimal (GTO) strategy for the OQA Information Game is equivalent to a greedy policy that, at each step, selects the query that maximizes the Expected Information Gain (EIG).*

*Sketch.* The GTO policy is found by solving the Bellman equation for the minimum expected future cost (number of queries). For the specific structure of the OQA game—uniform action costs and a uniform prior over candidates—the optimal cost-to-go function $C(S)$ for any belief state $S$ is a monotonically increasing function of the Shannon entropy $H(S)$. Therefore, the action that minimizes the expected future cost, $\mathbb{E}[C(S_o)]$, is the same action that minimizes the expected future entropy, $\mathbb{E}[H(S_o)]$. Minimizing expected future entropy is, by definition, equivalent to maximizing the one-step Expected Information Gain. Thus, the myopic EIG-maximizing policy and the globally optimal GTO policy coincide. A full proof is provided in §B.                                            □

This equivalence is powerful: it allows us to construct a perfect oracle that plays the GTO strategy simply by calculating the EIG for all possible questions at each turn and selecting the best one. This oracle provides the hard, information-theoretic lower bound for our benchmark.

## 3 AN ACTIVE INFERENCE AND BILEVEL OPTIMIZATION VIEW

Having defined the optimal strategy for the Information Game, we now frame it within the computational architecture of an ideal rational agent using the principles of active inference. Active inference unifies perception (belief updating) and action (decision making) under a single objective: minimizing variational free energy (Friston et al., 2017).

An active inference agent maintains a generative model of its world and seeks to minimize the free energy, which is an upper bound on surprise (negative log evidence). Perception is cast as an inner loop of updating beliefs to minimize free energy for a given observation. Action is cast as an outer loop of selecting policies that are expected to minimize free energy in the future.

**Theorem 3.1** (EFE Minimization equals EIG Maximization in OQA). *For the OQA Information Game, an agent whose policy is to minimize the Expected Free Energy (EFE) is implementing the same strategy as an agent that maximizes Expected Information Gain (EIG).*

*Sketch.* The Expected Free Energy is the expectation of the variational free energy over future outcomes. Under the game's assumptions (deterministic, noise-free observations), the EFE objective simplifies to minimizing the Shannon entropy of the predictive distribution over future answers, $H[p(o \mid \pi)]$. The EIG objective, defined as $H[p(x)] - \mathbb{E}_o[H[p(x \mid o)]]$, also simplifies to maximizing

this same quantity, $H[p(o \mid \pi)]$. Since both principles optimize the same mathematical quantity, the resulting policies are identical. The full derivation is provided in §C. □

This dual formulation of the optimal strategy—as both the GTO solution to a game and the policy of an ideal active inference agent—provides a deeply principled foundation for our benchmark. Furthermore, this separation of inference and control naturally defines a tractable bilevel optimization problem (Colson et al., 2007).

**Proposition 3.2** (Active Inference as Bilevel Optimization). *The active inference agent's task can be formulated as a differentiable bilevel optimization problem:*

$$\min_a \ \Phi(a) = \mathbb{E}_{o \sim p(\cdot \mid a)}\big[F\big(q^*(a, o)\big)\big] \quad s.t. \quad q^*(a, o) = \arg\min_q F(q; a, o). \tag{1}$$

*The outer loop optimizes the action parameters $a$ to minimize expected future free energy, while the inner loop solves for the optimal posterior belief $q^*$ given an action and a hypothetical outcome. This structure is amenable to gradient-based training methods. A full proof of differentiability is in Appendix §C*

This framework has direct implications for LLM alignment. If we model the user's intent as the latent state, then choosing a clarifying question becomes an EIG maximization problem. From this perspective, current RLHF pipelines (Askell et al., 2021; Bai et al., 2022) can be seen as approximate, ungrounded heuristics for solving this more formal bilevel objective.

## 4 BENCHMARK: OPTIMAL QUESTION ASKING IN LLMS

We measure how efficiently large language models (LLMs) gather information by comparing them with an *information-theoretic oracle* (see §D) across five binary-attribute guessing tasks:

- **Real-world:** PLACES, CARS, ANIMALS
- **Synthetic:** two corpora of 25 and 100 objects whose names are random hexadecimal keys and whose attributes are sampled uniformly at random

All attribute tables, the oracle, and the evaluation harness are released in our Supplementary Material.

**Dataset format.** Each object is represented by a Boolean attribute vector stored as JSON. Left: a snippet from ANIMALS. Right: a snippet from SYNTHETIC-25.

```
/* excerpt: 25-Animals */
{
  "cat":  {"mammal": true, "big": false, "can_swim": false, ...},
  "lion": {"mammal": true, "big": true,  "can_swim": false, ...}
}

/* excerpt: 25-Synthetic */
{
  "1a9f": {"a": false, "b": true,  "c": false, ...},
  "3b47": {"a": true,  "b": false, "c": true,  ...}
}
```

Each domain is provided in two sizes chosen so that (i) dialogs fit in a single LLM context window and (ii) the oracle remains tractable.

- **25-object tier** (lighter models): GPT-4o, Gemini 2.0 Flash, Claude 3.5 Haiku
- **100-object tier** (flagship models): GPT-o3, Gemini 2.5 Pro, Claude 3.7 Sonnet, Grok 3

Attribute vectors are unique in PLACES, CARS, and both synthetic sets, so an optimal agent can identify the target deterministically. ANIMALS deliberately includes duplicates, yielding *irreducible uncertainty* that tests whether a model can recognize ambiguity.

A hidden target is drawn uniformly. At every turn the agent asks a yes/no attribute question, receives a truthful answer, updates its posterior (implicitly, for LLMs), and queries again. The dialog ends when

only one candidate (or one equivalence class for ANIMALS) remains. External tools are disallowed for LLMs; the oracle is exempt because it defines the lower bound.

For each new target we start a fresh chat session with memory explicitly disabled, preventing cross-game context leakage. All interactions are conducted through the public chat interface rather than the API to match the default end-user setting.

Every model receives the generic prompt in Listing 1. Angle-bracket placeholders are instantiated per domain using Table 1. For each model, domain, and tier we run five random seeds and report the integer floor of the mean number of queries needed to isolate the target (lower is better). Table 2 also lists the oracle optimum (floored mean) for every tier.

```
Prompt (template)
-----------------
This is a <DATASET> attributes dataset. I have a hidden <OBJECT TYPE> in mind.
You are allowed to ask me binary questions about the hidden <OBJECT TYPE>'s
attributes, so you can figure out what it is. After receiving an answer,
at each step you must print your current belief distribution
<OPTIONAL: "and the calculated entropy drop"> about the possible hidden
<OBJECT TYPE>. We can stop when there are no more distinguishing attributes
between the remaining options. (You're not allowed to solve this using
programming.)
```

Figure 1: Generic prompt shown to every model.

| Domain | <DATASET> | <OBJECT TYPE> |
|---|---|---|
| Animals | animal attributes | animal |
| Cars | cars attributes | car |
| Places | places attributes | place |
| Synthetic | synthetic object attributes | object |

Table 1: Prompt-placeholder instantiations for the four domain families.

*Remark* 4.1 (Justification for Binary-Attribute Questions). The OQA game is intentionally constrained to binary (yes/no) questions about attributes. This simplification is crucial for establishing a tractable benchmark with a provably optimal solution. If the game were generalized to allow for more complex actions or payoffs, finding the optimal strategy would become computationally intractable. The general problem of finding a Nash Equilibrium is a famously hard problem, known to be *PPAD-complete* (Daskalakis et al., 2009). This means there is no known efficient (polynomial-time) algorithm to find the solution for the general case. By constraining the action space, we create a game structure where the GTO policy is computable, allowing us to build a perfect oracle and provide a true "gold standard" for evaluation.

We compare the LLMs against a perfect, Game-Theory Optimal player, which we implement as an information-theoretic oracle. This oracle plays by selecting the query that maximizes the **Expected Information Gain (EIG)** at each step, thereby minimizing the expected length of the game. The oracle's performance represents the information-theoretic lower bound on the number of questions required. A detailed breakdown of the oracle's algorithm and proof of optimality is provided in Appendix §D. Results on synthetic data are presented in Appendix §E

### 4.1 EXPERIMENTAL RESULTS

Table 2 reports the mean number of queries required by each model, while Figures 2–3 plot the corresponding entropy trajectories. Below we briefly highlight the main trends those curves reveal.

All models cut their posterior entropy by roughly half within the first three to four questions, echoing the oracle's near–binary-search behavior. Flagship systems (GPT-o3, Gemini 2.5 Pro, Claude 3.7 Sonnet) shadow the oracle most closely on PLACES and CARS, but stay about one extra query above it on the more ambiguous ANIMALS domain.

Table 2: Mean number of queries (lower is better). Values are the integer floor of the mean over five random targets per tier; oracle means differ between the 25- and 100-object settings.

| Model | 25-object tier | | | Model | 100-object tier | | |
|---|---|---|---|---|---|---|---|
| | Places | Cars | Animals | | Places | Cars | Animals |
| GPT-4o | 6 | 7 | 6 | GPT-o3 | 7 | 7 | 6 |
| Gemini 2.0 Flash | 6 | 6 | 5 | Gemini 2.5 Pro | 6 | 8 | 6 |
| Claude 3.5 Haiku | 7 | 6 | 6 | Claude 3.7 Sonnet | 5 | 7 | 8 |
| Oracle | 5 | 5 | 4 | Grok 3 | 7 | 7 | 7 |
| | | | | Oracle | 5 | 6 | 5 |

Lighter models (GPT-4o, Gemini 2.0 Flash, Claude 3.5 Haiku) close the gap to within two questions on the 25-object tier, yet their curves diverge from the oracle sooner than those of the flagship models once the search space shrinks below eight candidates. This suggests that larger models ask more optimal questions on average and maintain a near-optimal belief state even late in the dialogue.

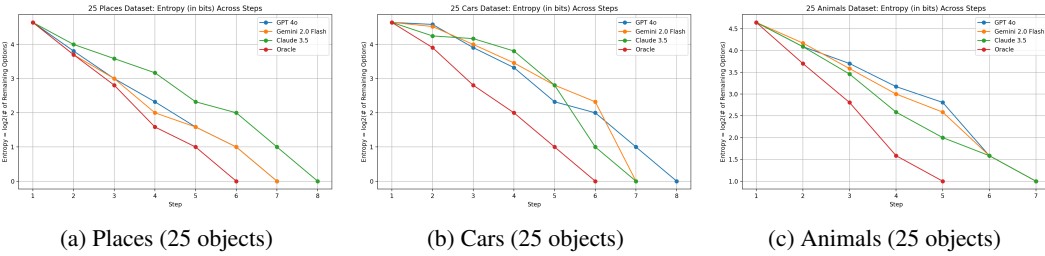

(a) Places (25 objects)  (b) Cars (25 objects)  (c) Animals (25 objects)

Figure 2: Posterior entropy (bits) versus dialog turn for lighter models on the 25-object tier. Each curve shows the integer floor of the mean over five random targets; lower curves indicate faster uncertainty reduction. The oracle curve marks the information-theoretic optimum.

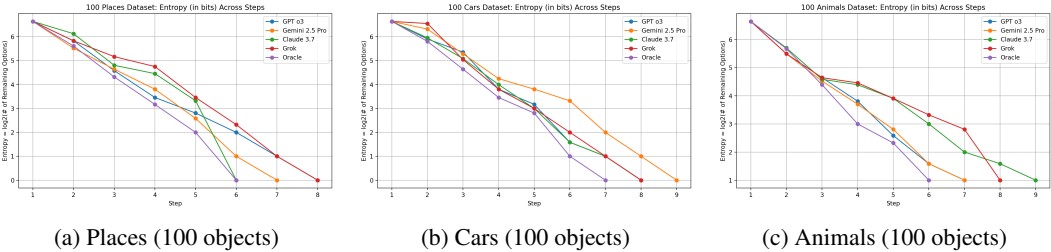

(a) Places (100 objects)  (b) Cars (100 objects)  (c) Animals (100 objects)

Figure 3: Posterior entropy (bits) versus dialog turn for flagship models on the 100-object tier. Each curve shows the integer floor of the mean over five random targets; lower curves indicate faster uncertainty reduction. The oracle curve is the information-theoretic optimum.

## 5 RELATED WORK

**Game theory and optimal strategy.** The fundamental premise of our work is that inquiry is a game of incomplete information. The solution concepts for such games originate with the Minimax Theorem for two-player, zero-sum games, which defines an unexploitable, or optimal, strategy (Von Neumann and Morgenstern, 1947). This was later generalized by the concept of the Nash Equilibrium for N-player and non-zero-sum games (Nash, 1951). We apply this game-theoretic lens to model information gathering, defining the optimal inquiry policy as the GTO strategy for resolving uncertainty.

**Evaluating agents in strategic environments.** The rigorous analysis of complex strategic behavior often relies on the study of simplified toy games." In poker theory, for instance, simple games like the *AKQ Game* are used to isolate and understand the core principles of bluffing and value betting

(Chen and Ankenman, 2006). This contrasts with many large-scale LLM benchmarks that test broad knowledge but lack a provably optimal solution, making it difficult to distinguish true strategic reasoning from heuristic pattern matching (Suzgun et al., 2022; Zellers et al., 2019). Our approach is more akin to benchmarks in multi-agent reinforcement learning (Zhang et al., 2021) , where agents are evaluated in well-defined environments, but with the distinct advantage of comparing against a computationally derived GTO oracle rather than just strong heuristics or prior agents. OQA is thus a toy game by design, created to provide a precise, quantitative measure of strategic efficiency that complements broader, qualitative benchmarks.

**Active inference.** The free-energy principle unifies perception and action as variational inference (Friston, 2010; Friston et al., 2017). Subsequent work introduces *epistemic value*, or information gain as an intrinsic drive, and surveys robotics and ML implementations (Parr and Friston, 2017; Lanillos et al., 2021; DaCosta et al., 2022). Recent studies extend the same framework to dialogue-based LLM agents (Shusterman et al., 2025) and to visual foraging in scene construction (Mirza et al., 2016). This unification of perception and action provides the formal objective for an optimal player in our Information Game; the agent's policy is to select queries that minimize expected free energy, thereby maximizing its long-term expected value by resolving uncertainty as efficiently as possible.

**Bilevel optimization.** Early surveys summarize the theory and classical applications of bilevel optimization (Bard, 1998; Colson et al., 2007). Casting hyperparameter tuning as a bilevel problem enables gradient-based search procedures (Pedregosa, 2016). Related formulations have been applied to meta-learning and neural-architecture search, where the outer loop optimizes validation loss while the inner loop updates model parameters (Franceschi et al., 2018; Ji et al., 2021; Liu et al., 2022). Bilevel formulations have also been explored for policy optimization in reinforcement learning (Chakraborty et al., 2023). In our work, this mathematical machinery operationalizes the active inference objective, allowing us to frame the search for a GTO policy as a solvable, differentiable problem where the outer loop selects strategic queries and the inner loop performs belief updates.

**Alignment via questioning.** Supervised or reinforcement-learning methods explicitly train LLMs to pose clarifying questions, including STaR-GATE (Andukuri et al., 2024) and Clarify-When-Necessary (Zhang and Choi, 2023). Related retrieval-augmented training improves knowledge-grounded dialogue by having the model issue iterative information queries to an external corpus (Shuster et al., 2021). Other work optimizes question generation by maximizing expected information gain (Mazzaccara et al., 2024). While these methods improve inquiry, they represent heuristic or exploitative strategies; OQA provides the first formal benchmark to measure the efficiency of such strategies against a provably optimal baseline.

**Planning in LLMs.** Chain-of-thought prompting (Wei et al., 2022) and Tree-of-Thought search (Yao et al., 2023a) show that show that structured prompting, and in the case of ToT, an explicit search over candidate thoughts, can elicit multi-step plans. ReAct interleaves these internal "thoughts" with web or tool calls (Yao et al., 2023b). Open-ended agents such as Voyager in MINECRAFT (Wang et al., 2023) and language-model–based zero-shot robotic planners (Huang et al., 2022) add hierarchical control that decomposes tasks into subgoals. These prompting and search techniques represent the internal cognitive architecture an agent uses to reason about the game state; our OQA benchmark provides a precise, external measure of the optimality of the resulting inquiry strategy.

## 6 LIMITATIONS AND DISCUSSION

Our study targets binary, closed-world tasks built from finite attribute tables, so OQA can overstate performance compared with real dialogs that need open-ended, scalar, or multimodal answers. A stricter benchmark would pair free-form query–answer sets with images, e.g., CLEVR-style attributes (Johnson et al., 2017). We also assume uniform target priors, yet real priors are often skewed, user specific, and time varying (Settles, 2009; Bayram et al., 2025); such priors change both the optimal policy and the model–oracle gap.

Our evaluation covers three real-world tables (Places, Cars, Animals) and two synthetic worlds (Appendix E) but omits temporal reasoning, vision-language queries, and embodied interaction (Mirza et al., 2016; Lanillos et al., 2021), which involve continuous state spaces and partial observability. Synthetic worlds remove linguistic cues and can scale arbitrarily, yet context-window limits hamper

current LLMs (Hosseini et al., 2024). Consequently, the entropy curves show "bumps" when forgotten candidates reappear after reminder prompts (Figures 4a–4b).

We deliberately disable retrieval, function calls, and scratchpads so we can isolate a model's intrinsic planning ability, even though production systems frequently embed such tools (Lewis et al., 2020; Schick et al., 2023; Nye et al., 2021). A complementary benchmark would help quantify how much of the overall planning load these tools actually absorb.

Under our setting, the oracle's dynamic program runs in $\mathcal{O}(d|S|^2)$ time and remains tractable for state spaces up to roughly 100 items; when tasks grow larger or move beyond binary decisions, exact solutions become impractical and one must rely on Monte Carlo sampling or deep-search oracles instead (Kirsch et al., 2019; Schrittwieser et al., 2020).

We also note that a model may look efficient because it recalls patterns learned during pretraining rather than planning in real time, so tests that shuffle labels or mask semantic cues are useful for teasing apart genuine reasoning from built-in bias (Wei et al., 2023; Shi and Penn, 2025). Meanwhile, language models generally need one to three additional questions on synthetic datasets, which exposes lingering weaknesses in counting and set manipulation (Yehudai et al., 2024; Dronen et al., 2024; Barbero et al., 2024); this synthetic–real gap is likely conservative, given the small scale of the synthetic tasks evaluated.

**Broader Impacts.** Asking shorter, more focused questions can lighten users' mental effort during AI-led tutoring sessions, literature screening, and in assistive-robot tasks. The same skill, however, can speed up privacy harvesting or persuasive targeting. Limiting the number of queries, applying user-level differential privacy, and giving people clear dashboard controls (Huang et al., 2024; Charles et al., 2024; Freiberger et al., 2025) can help reduce these risks to a certain extent.

OQA also ignores the cognitive friction that a machine's questions place on people. Even an entropy-optimal dialog can feel tedious if it repeats obvious attributes or violates social norms, reducing trust and engagement in ways observed in tutoring studies that track question quality and learner effort (Graesser and Person, 1994; Chi and Wylie, 2014). Adding a user-rated cost term or a simulated penalty for redundancy, latency, or awkward phrasing would push agents to trade a bit of information gain for a smoother conversation.

## 7 CONCLUSION AND FUTURE WORK

By framing active inference as a bilevel process, an outer loop that maximizes expected information gain and an inner loop that updates beliefs, we introduce Optimal Question Asking (OQA), a decision-theoretic benchmark that pairs an exact information-theoretic oracle with an automated harness to measure how quickly language agents reduce uncertainty, and experiments show that mid-tier LLMs need one to two more queries than the oracle on the 25-object tier while even flagship models require one to three extra queries on the 100-object tier, deficits that conventional accuracy metrics miss and that spotlight ongoing challenges in inquiry strategy and belief tracking.

Future work can include: (i) scaling the framework to thousands of items with continuous attributes and Bayesian or simulation-based oracles; (ii) adding multimodal inputs—images, audio, and structured tables—for vision-language and speech agents; (iii) evaluating planning when models can call external tools such as retrieval systems, calculators, and code executors (Yao et al., 2023b; Gao et al., 2023; Wen et al., 2024); and (iv) improving learning by testing bilevel gradient methods (Franceschi et al., 2018), preference-driven meta-learning (Piriyakulkij et al., 2023), and reinforcement learning to fine-tune question-efficient agents within OQA.

Because OQA is lightweight and fully deterministic, it can be embedded into continuous integration pipelines or reinforcement-learning loops to provide an on-policy signal for query efficiency. We also envision extensions where the oracle becomes a cooperative partner that teaches the model to trade off between expected information gain and auxiliary costs such as latency, token budget, or privacy leakage. Such multiobjective training could yield agents that not only know what to ask but also when and how to ask it, closing the gap between theoretical optimality and practical usability.

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

## REPRODUCIBILITY STATEMENT

All artifacts required to replicate our results are provided in a single ZIP file in the supplement.

- **Code.** Python 3.11 scripts for (i) the information-theoretic oracle, (ii) the automated evaluation loop, (iii) figure/table generation, and (iv) deterministic synthesis of the two synthetic corpora. Only NumPy and Matplotlib are required.
- **Data.** Eight JSON attribute tables—PLACES, CARS, ANIMALS, and SYNTHETIC at both the 25- and 100-object tiers—licensed under CC-BY-4.0.
- **Transcripts.** One complete sample dialog per model for every domain–tier pair: 25/100-PLACES, 25/100-CARS, 25/100-ANIMALS, and 25/100-SYNTHETIC.

We hope that releasing both the oracle and the evaluation harness will catalyze a community push toward query-aware benchmarks, much as GLUE (Wang et al., 2019) and BIG-bench (BIG-bench Collaboration, 2022) standardized answer-centric testing.

## A  A GAME-THEORETIC FORMULATION OF OPTIMAL INQUIRY

We posit that rational inquiry is not merely a task of passive information processing, but a dynamic, strategic game played under uncertainty. To analyze the efficiency of language agents in this domain, we first formalize the environment and the concept of an optimal strategy using the tools of game theory.

### A.1  MARKOV GAMES AND MULTI-STREET GAMES

We begin with the general definition of a Markov Game, which provides the formal structure for multi-agent sequential decision-making.

**Definition A.1** (Markov Game). A **Markov Game** (or stochastic game) is a tuple $\mathcal{M} = \langle \mathcal{P}, \mathcal{S}, \{\mathcal{A}_i\}_{i \in \mathcal{P}}, T, \{R_i\}_{i \in \mathcal{P}} \rangle$ where:

- $\mathcal{P} = \{1, ..., N\}$ is a finite set of $N$ players.

- $\mathcal{S}$ is a finite set of states.

- $\mathcal{A}_i$ is the finite set of actions available to player $i$. The joint action space is $\mathbf{A} = \times_{i \in \mathcal{P}} \mathcal{A}_i$.

- $T : \mathcal{S} \times \mathbf{A} \to \Delta(\mathcal{S})$ is the state transition function, where $\Delta(\mathcal{S})$ is the set of probability distributions over $\mathcal{S}$.

- $R_i : \mathcal{S} \times \mathbf{A} \to \mathbb{R}$ is the reward function for player $i$.

The game proceeds in discrete time steps. At each step $t$, the players observe the state $s_t \in \mathcal{S}$, simultaneously choose actions $a_{i,t} \in \mathcal{A}_i$, receive rewards $r_{i,t} = R_i(s_t, \mathbf{a}_t)$, and the state transitions to $s_{t+1} \sim T(s_t, \mathbf{a}_t)$.

Many real-world strategic interactions, including poker, can be modeled as \*\*multi-street games\*\*. These are sequential games where the decision-making process is divided into distinct stages or "streets," separated by stochastic events that change the game state.

**Example A.2** (The Game of Poker). No-limit Texas Hold'em is a canonical multi-street game. The game proceeds through up to four streets of betting (pre-flop, flop, turn, river), separated by the reveal of community cards. At each street, players make strategic decisions (bet, call, raise, fold) based on their private information (hole cards) and public information (board cards and opponent actions). The objective is to maximize expected winnings over the distribution of opponents' possible hands.

## B  A GAME-THEORETIC FORMULATION OF OPTIMAL INQUIRY

We posit that rational inquiry is not merely a task of passive information processing, but a dynamic, strategic game played under uncertainty. To analyze the efficiency of language agents in this domain, we first formalize the environment and the concept of an optimal strategy using the tools of game theory.

### B.1  MARKOV GAMES AND MULTI-STREET GAMES

We begin with the general mathematical structure for multi-agent sequential decision-making under uncertainty: the Markov Game. This framework provides the necessary formalism to model complex strategic interactions such as poker and, subsequently, our *Information Game*.

**Definition B.1** (Markov Game). A **Markov Game**, also known as a stochastic game, is a tuple $\mathcal{M} = \langle \mathcal{P}, \mathcal{S}, \{\mathcal{A}_i\}_{i \in \mathcal{P}}, T, \{R_i\}_{i \in \mathcal{P}} \rangle$ where:

- $\mathcal{P} = \{1, \ldots, N\}$ is a finite set of $N$ players.

- $\mathcal{S}$ is a finite set of world states.

- $\mathcal{A}_i$ is the finite set of actions available to player $i$. The joint action space is the Cartesian product $\mathbf{A} = \times_{i \in \mathcal{P}} \mathcal{A}_i$.

- $T : \mathcal{S} \times \mathbf{A} \to \Delta(\mathcal{S})$ is the state transition function, where $\Delta(\mathcal{S})$ denotes the space of probability distributions over $\mathcal{S}$.

- $R_i : \mathcal{S} \times \mathbf{A} \to \mathbb{R}$ is the reward function for player $i$.

The game proceeds in discrete time steps $t = 0, 1, 2, \dots$. At each step $t$, the players observe the current state $s_t \in \mathcal{S}$, simultaneously choose actions $a_{i,t} \in \mathcal{A}_i$ to form a joint action $\mathbf{a}_t \in \mathbf{A}$, receive individual rewards $r_{i,t} = R_i(s_t, \mathbf{a}_t)$, and the system transitions to a new state $s_{t+1} \sim T(s_t, \mathbf{a}_t)$. In games of *incomplete information*, the state $s_t$ is not fully observable to all players.

Many real-world strategic interactions, particularly those in card games, can be modeled as *multi-street games*, a specific and important class of Markov game.

**Definition B.2** (Multi-Street Game). A **Multi-Street Game** is a Markov Game whose temporal structure is partitioned into a finite sequence of *streets*, $j \in \{1, \dots, J\}$.

- Within each street $j$, a sequence of intra-street time steps occurs where the state transitions are determined solely by the players' joint actions.

- The transition from street $j$ to street $j + 1$ is governed by a stochastic chance move, the outcome of which is independent of the players' actions in street $j$.

This structure is the defining characteristic of modern variants of poker.

**Example B.3** (No-Limit Texas Hold'em as a Multi-Street Game). The game of No-Limit Texas Hold'em is a canonical instance of a multi-street game.

- The game consists of up to four streets ($J = 4$): Pre-flop, Flop, Turn, and River.

- The transition between streets is a stochastic chance move: the dealing of public community cards, drawn from the remaining deck.

- Within each street, a structured sequence of betting actions (e.g., check, bet, call, raise, fold) takes place. The state transitions within a street are deterministic, conditioned on the players' actions.

- The state space $\mathcal{S}$ is vast, comprising the public board state, the history of actions, player stack sizes, and each player's private hole cards. The private component of the state introduces incomplete information.

**Proposition B.4.** *No-Limit Texas Hold'em is a finite, multi-player, zero-sum (ignoring rake), sequential Markov game of incomplete information with a multi-street structure.*

*Proof.* The set of players is finite. The set of all possible card distributions and betting sequences is finite, hence the state space $\mathcal{S}$ is finite, albeit exceptionally large. The action set $\mathcal{A}_i$ at any decision point is finite. State transitions are governed by player actions and chance moves, consistent with the definition of a Markov Game. The total monetary exchange sums to zero (in a cash game setting excluding rake). The sequential nature of actions and players' private holdings constitute a sequential game of incomplete information. It therefore conforms to the specified class of game. $\qquad \square$

*Remark* B.5. This formalism is critical. It establishes that complex strategic environments like poker—and, as we will show, the OQA Information Game—are not merely ad-hoc scenarios but are instances of a well-defined mathematical object. The solution concepts developed for Markov games, namely the Nash Equilibrium, are therefore the correct and rigorous tools for their analysis.

### B.2 THE AKQ GAME: A CANONICAL TOY GAME FOR STRATEGIC ANALYSIS

The complexity of real-world games, such as poker, with their vast state and action spaces, renders the direct analysis of their optimal strategies computationally intractable. A standard and powerful methodology in game theory is therefore to analyze simplified, abstract models known as *toy games* These games are constructed to be simple enough to be solvable, yet rich enough to isolate and reveal fundamental strategic principles that generalize to their more complex counterparts. The most famous of these in the domain of poker is the AKQ game. Its formal analysis is non-trivial and serves as a quintessential illustration of bluffing, value betting, and the crucial concept of strategic indifference.

**Definition B.6** (The AKQ Game ([Chen and Ankenman, 2006](#))). The AKQ game is a two-player, zero-sum, sequential game of incomplete information defined by the following extensive form:

(i) **Players and Deck:** The set of players is $\mathcal{P} = \{1, 2\}$. The deck is $\mathcal{C} = \{A, K, Q\}$, with the ordinal ranking $A \succ K \succ Q$.

(ii) **Initial Node (Chance Move):** Nature deals one card to each player without replacement from $\mathcal{C}$. Each of the $3 \times 2 = 6$ possible deals is equally likely. Each player contributes an ante of 1 unit to an initial pot of size $P = 2$.

(iii) **Actions and Subgames:** Player 1 acts first from the action set $\mathcal{A}_1 = \{\text{Bet}, \text{Check}\}$. A bet is for a fixed size of 1 unit.

- If Player 1 bets, Player 2 responds from $\mathcal{A}'_2 = \{\text{Call}, \text{Fold}\}$.
- If Player 1 checks, Player 2 responds from $\mathcal{A}_2 = \{\text{Bet}, \text{Check}\}$. If Player 2 bets, Player 1 responds from $\mathcal{A}'_1 = \{\text{Call}, \text{Fold}\}$. If Player 2 checks, the game terminates.

(iv) **Terminal Nodes and Payoffs:** The game terminates upon a fold or a final check/call. At showdown, the player with the higher-ranking card wins the pot. The utility for a player is their net gain or loss for the hand.

### B.2.1 STRATEGIC REDUCTION VIA ITERATED ELIMINATION OF DOMINATED STRATEGIES

The solution process begins by simplifying the strategy space. A strategy is strictly dominated if another strategy yields a strictly higher payoff for every possible strategy of the opponent.

**Lemma B.7** (Dominated Strategies). *The following strategies are strictly dominated and can be eliminated from the set of rationalizable strategies:*

*(a) For Player 1: With an Ace, check-folding; with a Queen, check-calling.*

*(b) For Player 2: With an Ace, checking after a check; with a King, betting after a check.*

*Proof.* The proof proceeds by direct inspection of payoffs at the relevant decision nodes.

- If Player 1 holds an Ace, they are guaranteed to win at showdown, making calling always superior to folding. If Player 1 holds a Queen and faces a bet from Player 2 (who must hold A or K), Player 1 is guaranteed to lose at showdown, making folding superior to calling.

- If Player 2 holds an Ace and Player 1 checks, betting can induce a call from a worse hand (King), making betting strictly better than checking. If Player 2 holds a King and Player 1 checks, betting will only be called by a better hand (Ace) and will fold a worse hand (Queen). The expected utility of betting is thus negative, while checking is non-negative.

$\square$

*Remark* B.8. The elimination of dominated strategies reveals the core strategic tensions of the game. The optimal strategies must resolve: (1) Player 1's decision to bluff with a Queen; (2) Player 1's decision to call with a King (a "bluff-catcher"); and (3) Player 2's decision to bluff with a Queen.

### B.2.2 THE MIXED-STRATEGY EQUILIBRIUM SOLUTION

After eliminating dominated strategies, neither player has a pure strategy that is optimal for all situations. For example, if Player 1 *always* bluffs with a Queen, Player 2 can exploit this by always calling with a King. If Player 1 *never* bluffs, Player 2 can exploit this by folding a King to any bet. The solution must therefore be a **mixed strategy**, where players randomize their actions with specific frequencies.

The equilibrium is found by applying the **Principle of Indifference**: each player must mix their strategies in such a way that the opposing player is made indifferent between two of their own actions.

**Theorem B.9** (GTO Strategy for the AKQ Game). *The Game-Theory Optimal strategy profile involves the following mixed strategies:*

- *Player 1's Strategy:*

    - *With an Ace: Always bet.*
    - *With a King: Always check. If Player 2 bets, call.*
    - *With a Queen: Bet (bluff) with probability $\frac{1}{3}$, and check with probability $\frac{2}{3}$. If checked and Player 2 bets, fold.*

- *Player 2's Strategy:*

    - *When facing a bet from Player 1:*
        * *With an Ace: Always call.*
        * *With a King: Call (bluff-catch) with probability $\frac{1}{3}$, and fold with probability $\frac{2}{3}$.*
        * *With a Queen: Always fold.*
    - *After Player 1 checks:*
        * *With an Ace: Always bet.*
        * *With a King: Always check.*
        * *With a Queen: Always bet (bluff).*

*Proof by Indifference.* The bluffing frequency for Player 1 with a Queen is chosen to make Player 2 indifferent between calling and folding with a King. When Player 1 bets, the pot is 3 (2 antes + 1 bet). Player 2 must call 1 to win 3. To make this call break-even, Player 2 must believe their probability of winning is $\frac{1}{1+3} = \frac{1}{4}$. Player 1's betting range consists of all Aces (1 combination) and Queens (bluffed with probability $b$). The ratio of bluffs to total bets must be $\frac{b}{1+b} = \frac{1}{4}$, which solves to $b = \frac{1}{3}$. A similar indifference calculation for Player 1 determines Player 2's optimal calling frequency with a King. $\qquad\square$

*Remark* B.10. The solution to the AKQ game demonstrates that optimal strategic play is not about finding a single "best" move, but about constructing a balanced, unexploitable strategy that correctly mixes actions. This is the essence of a GTO solution.

## B.3 THE GAME-THEORETIC SOLUTION CONCEPT: SUBGAME PERFECT NASH EQUILIBRIUM

Having established the utility of toy games like AKQ, we now formalize the tools required to solve them. The foundational solution concept in game theory is the Nash Equilibrium.

**Definition B.11** (Nash Equilibrium (Nash, 1951)). A strategy profile (a set of strategies for all players) is a **Nash Equilibrium** if no player can achieve a better outcome by unilaterally changing their own strategy, assuming all other players' strategies remain unchanged. In a two-player, zero-sum game, a Nash Equilibrium strategy is considered **Game-Theory Optimal (GTO)** as it is unexploitable.

For sequential games, the game's structure is captured by an **extensive-form game tree**. In such games, a stronger equilibrium concept is required to rule out non-credible threats.

**Definition B.12** (Subgame Perfect Nash Equilibrium (SPNE)). A strategy profile is a **Subgame Perfect Nash Equilibrium** if it constitutes a Nash Equilibrium not only for the entire game but also for every possible **subgame** within it. A subgame is a part of the game tree that starts at a single decision node and contains all subsequent nodes.

### B.3.1 FINDING THE SPNE VIA BACKWARD INDUCTION AND THE BELLMAN EQUATION

For any finite game of perfect information (where all players know the complete history of all previous actions), the SPNE can be found via a recursive algorithm known as **backward induction** (Maschler et al., 2020). The process begins at the terminal nodes (leaves) of the game tree and works backward towards the root.

This recursive logic is formally captured by **Bellman's principle of optimality**. Let $V^*(s)$ be the optimal value (maximum expected utility) of being in a state (or node) $s$. The principle states that an optimal policy has the property that whatever the initial state and initial decision are, the remaining

decisions must constitute an optimal policy with regard to the state resulting from the first decision. This gives rise to the **Bellman Equation** for the optimal value function:

$$V^*(s) = \max_{a \in \mathcal{A}(s)} \left( R(s,a) + \gamma \sum_{s' \in \mathcal{S}} T(s'|s,a)V^*(s') \right)$$

where $R(s,a)$ is the immediate reward from taking action $a$ in state $s$, $T(s'|s,a)$ is the transition probability to state $s'$, and $\gamma$ is a discount factor. For a finite, undiscounted, deterministic game, this equation provides the mechanism to solve for the optimal action at each node by recursively considering the optimal values of the subsequent nodes.

### B.4 The LLM Information Game: A Solvable Model of Inquiry

We now apply this formal machinery to define our central object of study: the **Information Game**. This serves as a formal toy game for the strategic problem of sequential information gathering.

**Definition B.13** (The OQA Information Game). The **OQA Information Game**, denoted $\mathcal{G}_{OQA}$, is a two-player, zero-sum, sequential game specified by the tuple $\langle \mathcal{P}, \mathcal{X}, \mathcal{S}, \{\mathcal{A}_i\}, T, \{R_i\} \rangle$, where:

(i) **Players:** $\mathcal{P} = \{I, R\}$, the *Inquirer* and the *Responder*.

(ii) **Hidden State Space** ($\mathcal{X}$)**:** $\mathcal{X}$ is a finite set of target objects. Nature makes an initial chance move, selecting a target $x^* \in \mathcal{X}$ according to a uniform prior distribution. This target $x^*$ is revealed only to the Responder.

(iii) **Public State Space** ($\mathcal{S}$)**:** A state $s_t \in \mathcal{S}$ is the set of candidate objects currently considered possible by the Inquirer, $S_t \subseteq \mathcal{X}$. The initial state is $S_0 = \mathcal{X}$.

(iv) **Action Spaces** ($\mathcal{A}_I, \mathcal{A}_R$)**:** At a non-terminal state $S_t$ (where $|S_t| > 1$), the Inquirer chooses an action $a_t \in \mathcal{A}_I(S_t)$, where $\mathcal{A}_I(S_t)$ consists of all binary attributes that produce a non-trivial partition of $S_t$. The Responder's action space $\mathcal{A}_R$ is $\{\text{yes}, \text{no}\}$. The Responder's action $o_t$ is a deterministic function of $a_t$ and $x^*$.

(v) **Transition Function** ($T$)**:** The transition between public states is deterministic. Given state $S_t$, Inquirer's action $a_t$, and Responder's answer $o_t$, the subsequent state is $S_{t+1} = \{s \in S_t \mid \text{attribute } a_t(s) \text{ corresponds to } o_t\}$.

(vi) **Reward Function** ($R_I, R_R$)**:** For each action taken by the Inquirer, it receives a reward $R_I = -1$. As the game is zero-sum, $R_R = -R_I = 1$. The game terminates when $|S_t| \leq 1$.

*Remark* B.14 (Perfect vs. Incomplete Information). It is crucial to distinguish the nature of information in this game. The underlying problem for the Inquirer is one of *incomplete information*, as the true target $x^*$ is unknown. However, the game as played over the sequence of public states $\{S_t\}$ is one of *perfect information*. The current candidate set $S_t$ is public knowledge, and there is no private information regarding the structure of the game tree itself. This structure allows the game's equilibrium to be solved via backward induction.

**Proposition B.15.** *The state graph of the OQA Information Game is a Directed Acyclic Graph (DAG).*

*Proof.* Let $S_t$ be a state. Any action $a_t \in \mathcal{A}_I(S_t)$ is, by definition, a query that partitions $S_t$ into at least two non-empty subsets, $S_{t,\text{yes}}$ and $S_{t,\text{no}}$. The subsequent state $S_{t+1}$ will be one of these subsets. Therefore, $|S_{t+1}| < |S_t|$. Since the size of the state set strictly decreases with every transition, the game cannot revisit a state and must terminate in a finite number of steps. The game is therefore acyclic. $\square$

### B.5 Solving the OQA Game: The GTO Policy via Backward Induction

Given that $\mathcal{G}_{OQA}$ is a finite, sequential game of perfect information, its Subgame Perfect Nash Equilibrium can be computed using the principle of backward induction. Let $C(S)$ be the minimum expected cost (number of future queries) for the Inquirer starting from a state (candidate set) $S$. This

cost function is the negative of the Inquirer's optimal value function, $C(S) = -V^*(S)$. The Bellman equation, reframed as a cost-minimization problem, is:

$$C(S) = 1 + \min_{a \in \mathcal{A}(S)} \mathbb{E}_{o \sim p(o|S,a)}[C(S_o)]$$

with the boundary condition $C(S) = 0$ for $|S| \leq 1$. The term $p(o|S, a)$ is the probability of observing answer $o$ given the query $a$ and current set $S$. Under a uniform prior over the elements of $S$, this is simply $|S_o|/|S|$. The equation expands to:

$$C(S) = 1 + \min_{a \in \mathcal{A}(S)} \left[ \frac{|S_{\text{yes}}|}{|S|} C(S_{\text{yes}}) + \frac{|S_{\text{no}}|}{|S|} C(S_{\text{no}}) \right]$$

The Game-Theory Optimal policy, $\pi^*$, is to select the action $a$ that achieves this minimum at every state $S$. The dynamic programming algorithm used for our oracle is a direct memoized implementation of this recursion, which guarantees it computes the SPNE for the OQA game.

## B.6 EQUIVALENCE OF THE GTO POLICY AND MAXIMIZING INFORMATION GAIN

The final and most critical step is to demonstrate that the multi-step optimal policy $\pi^*$ derived from backward induction is equivalent to the seemingly myopic (one-step greedy) policy of maximizing the **Expected Information Gain (EIG)** at each step. This equivalence justifies using EIG as the decision criterion for the optimal oracle.

The framework of **Bayesian Experimental Design (BED)** is concerned with choosing designs (experiments or queries) to "optimally" gather data (Rainforth et al., 2024). The most common objective in BED is to maximize the EIG, defined as the expected reduction in Shannon entropy from the prior to the posterior distribution.

$$\text{EIG}(\xi; p(\theta)) = \mathbb{E}_{y|\xi}[H[p(\theta)] - H[p(\theta|y, \xi)]]$$

In our context, the design $\xi$ is the query $a$, the outcome $y$ is the answer $o$, and the parameter $\theta$ is the hidden object $x^*$. With a uniform prior over the candidates in $S$, the EIG of a query $a$ simplifies to:

$$\text{EIG}(a; S) = H(S) - \left( \frac{|S_{\text{yes}}|}{|S|} H(S_{\text{yes}}) + \frac{|S_{\text{no}}|}{|S|} H(S_{\text{no}}) \right)$$

where $H(S) = \log_2(|S|)$. Maximizing EIG is equivalent to minimizing the expected posterior entropy, $\mathbb{E}_{o \sim p(o|S,a)}[H(S_o)]$.

While for general sequential design problems a greedy EIG policy can be sub-optimal (Rainforth et al., 2024), we prove it is globally optimal for the specific structure of the OQA game.

**Theorem B.16** (Equivalence of GTO and EIG Maximization in the OQA Game). *In the OQA Information Game, the GTO policy $\pi^*$ that solves the Bellman equation for minimum expected cost is equivalent to the greedy policy $\pi_{EIG}$ that selects the query maximizing the EIG at each step.*

*Proof.* The proof rests on the fact that the optimal cost $C(S)$ is a monotonic function of the entropy $H(S)$.

1. **Cost depends only on set size:** From the recursive structure of the optimal cost function $C(S)$, the cost of any subproblem, $C(S')$, depends only on the properties of the set $S'$, not on the path taken to reach it. In our game, with uniform action costs and a uniform prior, the optimal cost $C(S')$ is determined solely by the cardinality of the set, $|S'|$. That is, $C(S) = f(|S|)$ for some function $f$.

2. **Monotonicity:** It is self-evident that if $|S_1| > |S_2|$, then $C(S_1) \geq C(S_2)$, since at least as many queries will be required on average to resolve a larger set of possibilities. Thus, $C(S)$ is a monotonically increasing function of $|S|$.

3. **Entropy as a proxy for cost:** The Shannon entropy, $H(S) = \log_2(|S|)$, is also a strictly monotonically increasing function of $|S|$. Therefore, the optimal cost $C(S)$ must be a monotonic function of the entropy $H(S)$.

4. **Connecting the objectives:**

   - The **GTO policy** aims to find the action $a^*$ that solves: $\min_a \mathbb{E}_o[C(S_o)]$.
   - The **EIG policy** aims to find the action $a^{**}$ that solves: $\min_a \mathbb{E}_o[H(S_o)]$.

5. Since $C(\cdot)$ is a monotonic function of $H(\cdot)$, the action $a$ that minimizes the expected value of one will also minimize the expected value of the other. The preference ordering over actions induced by the expected future cost is identical to the preference ordering induced by the expected future entropy.

Therefore, the greedy, myopic policy of maximizing one-step EIG is identical to the globally optimal GTO policy found via multi-step backward induction. This proves that our oracle, which maximizes EIG at each decision node, is an exact implementation of the Subgame Perfect Nash Equilibrium of the OQA game. $\square$

*Remark* B.17 (From Game Theory to Agent Architecture). The preceding sections have established the formal game-theoretic structure of optimal inquiry. We defined the OQA Information Game, identified its solution concept as the Subgame Perfect Nash Equilibrium (SPNE), and proved that this GTO strategy is equivalent to greedily maximizing Expected Information Gain.

This raises a crucial question: What is the computational architecture of an agent that can successfully *implement* this optimal strategy? An agent cannot simply be handed a pre-computed game tree; it must possess an internal, principled mechanism for perception (updating its beliefs about the hidden state) and action (selecting the next query).

The following section addresses this by introducing the framework of **active inference**. We will demonstrate that an agent operating under the principles of active inference naturally implements the GTO strategy. Active inference provides the probabilistic and decision-theoretic machinery that allows an agent to play the Information Game optimally by unifying belief-updating and action-selection under a single objective: the minimization of free energy. This provides a powerful, first-principles account of the cognitive processes required for rational inquiry.

# C  ACTIVE INFERENCE, BILEVEL OPTIMIZATION, AND LLM ALIGNMENT

This section provides the formal probabilistic and decision-theoretic underpinnings of the optimal inquiry agent, framed within the principles of active inference. Active inference posits that a rational agent's behavior, encompassing both perception and action, can be cast as a process of minimizing a single objective: variational free energy. We develop this principle in three stages: first, by defining the perception/inference task as the minimization of variational free energy; second, by defining the action-selection task as the minimization of *expected* future free energy; and third, by showing how this entire process forms a tractable bilevel optimization problem.

## C.1  THE GENERATIVE MODEL AND MODELING ASSUMPTIONS

An active inference agent operates using a generative model of its environment, $p(o, x \mid \pi)$, which specifies the joint probability of observations $o$ and their latent causes (hidden states) $x$, conditioned on the agent's policy $\pi$ (a sequence of actions). For the OQA game, we make two critical simplifying assumptions:

1. **Uniform Prior over Latent States:** Before any observations are made, the agent's belief about the hidden state $x$ is a uniform distribution, $p(x) = \mathcal{U}(x)$.

2. **Deterministic, Noise-Free Observations:** The observation model $p(o \mid x, \pi)$ is deterministic. For a given hidden state $x$ and query (action) $\pi$, the resulting observation (answer) $o$ is uniquely determined. Consequently, the probability mass is concentrated on a single outcome: $p(o \mid x, \pi) \in \{0, 1\}$.

As we will show, under these conditions, the objective for action selection simplifies from minimizing expected free energy to maximizing expected information gain.

## C.2 Perception as Variational Free Energy Minimization

The first task for an agent is perception: inferring the hidden causes $x$ of its sensory observations $o$. This requires computing the posterior distribution $p(x \mid o, \pi)$. In many realistic scenarios, this posterior is computationally intractable. Variational inference addresses this by introducing a tractable, parametric family of distributions, $q(x; \phi)$, and then finding the member of this family that is closest to the true posterior. This is achieved by minimizing the **variational free energy**, $F(q)$.

**Definition C.1** (Variational Free Energy). The variational free energy $F(q)$ is defined as the Kullback-Leibler (KL) divergence between the approximate posterior $q(x)$ and the generative model's joint distribution $p(o, x \mid \pi)$:

$$F(q) := D_{\mathrm{KL}}\big[q(x) \parallel p(o, x \mid \pi)\big]$$
$$= \int q(x) \log \frac{q(x)}{p(o, x \mid \pi)} dx$$
$$= \mathbb{E}_{q(x)}\big[\ln q(x) - \ln p(o, x \mid \pi)\big].$$

By rearranging terms, we can see that free energy provides an upper bound on the negative log evidence (or "surprise"), $-\ln p(o \mid \pi)$:

$$F(q) = \mathbb{E}_{q(x)}\big[\ln q(x) - \ln p(x \mid o, \pi) - \ln p(o \mid \pi)\big]$$
$$= \mathbb{E}_{q(x)}\big[\ln \frac{q(x)}{p(x \mid o, \pi)}\big] - \ln p(o \mid \pi)$$
$$= D_{\mathrm{KL}}\big[q(x) \parallel p(x \mid o, \pi)\big] - \ln p(o \mid \pi). \tag{2}$$

Since the KL divergence is non-negative, $F(q) \geq -\ln p(o \mid \pi)$. Therefore, minimizing the free energy with respect to $q$ simultaneously (i) minimizes the discrepancy between the approximate and true posterior, and (ii) tightens the bound on the (negative) model evidence. The optimal posterior, $q^*(x) = \arg\min_q F(q)$, is the true posterior $p(x \mid o, \pi)$, at which point the KL divergence term becomes zero.

## C.3 Action Selection as Expected Free Energy Minimization

Active inference frames action selection as a process of choosing policies $\pi$ that are expected to minimize the free energy of the future. The agent evaluates each potential policy by calculating the **Expected Free Energy (EFE)**, denoted $\mathcal{G}(\pi)$.

**Definition C.2** (Expected Free Energy). The Expected Free Energy of a policy $\pi$ is the free energy expected over future outcomes $o$ that would be generated under that policy:

$$\mathcal{G}(\pi) = \mathbb{E}_{o \sim p(o \mid \pi)}\big[F\big(q^*(o, \pi)\big)\big], \tag{3}$$

where $q^*(o, \pi) = \arg\min_q F(q)$ is the optimal posterior belief the agent would hold *after* having executed policy $\pi$ and observed outcome $o$. The optimal policy $\pi^*$ is the one that minimizes this expectation:

$$\pi^* = \arg\min_\pi \mathcal{G}(\pi).$$

**Proposition C.3** (EFE equals negative expected information gain). *Under the assumptions of a uniform prior and deterministic observations, minimizing the expected free energy $\mathcal{G}(\pi)$ is equivalent to maximizing the mutual information $I_\pi[x; o]$ (i.e., the expected information gain) between the latent state $x$ and future observations $o$.*

*Proof.* The proof proceeds by showing that both quantities reduce to the negative entropy of the marginal distribution over future outcomes, $-H[p(o \mid \pi)]$.

**Step 1: Simplify the EFE.** For any given future outcome $o$, the optimal posterior is the true posterior, $q^*(o, \pi) = p(x \mid o, \pi)$. Substituting this into the definition of free energy from Eq. (2), the KL divergence term vanishes:

$$F\big(q^*(o, \pi)\big) = D_{\mathrm{KL}}\big[p(x \mid o, \pi) \parallel p(x \mid o, \pi)\big] - \ln p(o \mid \pi) = -\ln p(o \mid \pi).$$

Now, substituting this into the definition of EFE from Eq. (3):

$$\mathcal{G}(\pi) = \mathbb{E}_{o \sim p(o|\pi)}[-\ln p(o \mid \pi)] = H[p(o \mid \pi)].$$

Thus, minimizing EFE is equivalent to minimizing the entropy of the distribution over future outcomes.

**Step 2: Simplify the Mutual Information.** The mutual information (or expected information gain) is defined as:

$$I_\pi[x; o] = H[p(x)] - H[p(x \mid o, \pi)].$$

Alternatively, it can be written as $I_\pi[x; o] = H[p(o \mid \pi)] - H[p(o \mid x, \pi)]$. Under our assumptions:

- $H[p(o \mid x, \pi)]$ is the conditional entropy of outcomes given the latent state. Since the observation model $p(o \mid x, \pi)$ is deterministic, knowing $x$ removes all uncertainty about $o$. Therefore, $H[p(o \mid x, \pi)] = 0$.

This leaves us with:

$$I_\pi[x; o] = H[p(o \mid \pi)] - 0 = H[p(o \mid \pi)].$$

**Step 3: Equate the objectives.** From Step 1, minimizing $\mathcal{G}(\pi)$ is equivalent to minimizing $H[p(o \mid \pi)]$. From Step 2, maximizing $I_\pi[x; o]$ is equivalent to maximizing $H[p(o \mid \pi)]$. Therefore, minimizing the Expected Free Energy is equivalent to maximizing the Expected Information Gain. $\square$

## C.4 ACTIVE INFERENCE AS A DIFFERENTIABLE BILEVEL OPTIMIZATION PROBLEM

The separation of perception (inference) and control (action-selection) naturally gives rise to a bilevel optimization structure (Colson et al., 2007).

**Definition C.4** (Bilevel Optimization Formulation)**.** The active inference agent's problem can be formulated as:

$$\min_a \ \Phi(a) = \mathbb{E}_{o \sim p(\cdot|a)}\big[F\big(q^*(a, o)\big)\big] \quad \text{subject to} \quad q^*(a, o) = \arg\min_q F(q; a, o). \tag{4}$$

Here, the action $a$ is equivalent to the policy $\pi$.

- The **Outer Problem** is to select an action $a$ that minimizes the outer objective $\Phi(a)$, which is the Expected Free Energy.

- The **Inner Problem** is to find the optimal posterior belief $q^*$ for a *given* action $a$ and a *hypothetical* future observation $o$. The solution to the inner problem, $q^*(a, o)$, is a required input to evaluate the outer objective.

**Proposition C.5** (Differentiability)**.** *If the generative model $p(o, x \mid a)$ is differentiable with respect to the action parameters $a$, and the approximate posterior $q(x; \phi)$ is differentiable with respect to its parameters $\phi$, then Equation (4) defines a differentiable bilevel problem. This structure makes the problem accessible to modern gradient-based bilevel optimization solvers.*

*Proof.* The goal is to demonstrate that the gradient of the outer objective, $\nabla_a \Phi(a)$, exists and is computable. The outer objective is:

$$\Phi(a) = \mathbb{E}_{o \sim p(\cdot|a)}\big[F\big(q^*(a, o)\big)\big]$$

where $q^*(a, o) = q(x; \phi^*(a, o))$ and $\phi^*(a, o) = \arg\min_\phi F(\phi; a, o)$. The inner objective is $F(\phi; a, o) = \mathbb{E}_{q(x;\phi)}\big[\ln q(x; \phi) - \ln p(o, x \mid a)\big]$.

The proof proceeds in three main steps: (1) handling the expectation with respect to $a$, (2) differentiating the inner term using the chain rule, and (3) computing the necessary Jacobian using the implicit function theorem.

**Step 1: Differentiating the Expectation.** The gradient operator must be applied to an expectation where the distribution itself depends on the parameter $a$. We can address this using the reparameterization trick, assuming a suitable generative process for the observations $o$. Let's assume we can express the sampling of $o$ as a deterministic and differentiable function $g$ of the parameters $a$ and a random noise variable $\epsilon$, where $\epsilon \sim p(\epsilon)$ and its distribution does not depend on $a$. That is, $o = g(\epsilon, a)$. This allows us to rewrite the expectation:

$$\Phi(a) = \mathbb{E}_{\epsilon \sim p(\epsilon)}\big[F\big(q^*(a, g(\epsilon, a))\big)\big]$$

Now, the expectation is over a fixed distribution, so we can move the gradient operator inside:

$$\nabla_a \Phi(a) = \mathbb{E}_{\epsilon \sim p(\epsilon)}\big[\nabla_a F\big(q^*(a, g(\epsilon, a))\big)\big] \tag{5}$$

We now focus on computing the term inside the expectation for a fixed observation $o = g(\epsilon, a)$.

**Step 2: Applying the Chain Rule to the Inner Term.** The term $F(q^*(a, o))$ depends on $a$ in two ways: directly through the generative model $p(o, x \mid a)$ within the definition of $F$, and indirectly through the optimal inner parameters $\phi^*(a, o)$ which depend on $a$. Let $F$ be shorthand for $F(\phi; a, o)$. Using the multivariate chain rule, the total derivative of $F$ with respect to $a$ at the optimum $\phi^*$ is:

$$\frac{dF(\phi^*(a, o); a, o)}{da} = \left.\frac{\partial F}{\partial a}\right|_{\phi=\phi^*} + \left.\frac{\partial F}{\partial \phi^T}\right|_{\phi=\phi^*} \frac{\partial \phi^*}{\partial a} \tag{6}$$

By definition, $\phi^*$ is the minimizer of the inner objective $F$. Therefore, the first-order optimality condition holds:

$$\nabla_\phi F(\phi; a, o)|_{\phi=\phi^*} = 0$$

This means the second term in Equation (6) vanishes: $\left.\frac{\partial F}{\partial \phi^T}\right|_{\phi=\phi^*} = \mathbf{0}^T$. This is a crucial simplification known as the envelope theorem. The total derivative simplifies to just the partial derivative:

$$\frac{dF(\phi^*(a, o); a, o)}{da} = \left.\frac{\partial F}{\partial a}\right|_{\phi=\phi^*}$$

However, in practice, the inner problem is solved iteratively, and $\phi$ may only be an approximation of $\phi^*$. For a fully general and robust gradient computation, we must compute the Jacobian $\frac{\partial \phi^*}{\partial a}$. We proceed with this more general case.

**Step 3: Computing the Jacobian via the Implicit Function Theorem.** The optimal parameters $\phi^*(a, o)$ are defined implicitly by the first-order optimality condition:

$$G(\phi, a) := \nabla_\phi F(\phi; a, o) = 0$$

This equation holds at $\phi = \phi^*(a, o)$. The Implicit Function Theorem states that if we have an equation $G(\phi, a) = 0$ that implicitly defines $\phi$ as a function of $a$, then the Jacobian of $\phi$ with respect to $a$ is given by:

$$\frac{\partial \phi^*}{\partial a} = -\left[\nabla_\phi G(\phi^*, a)\right]^{-1}\left[\nabla_a G(\phi^*, a)\right]$$

Let's identify the terms in our context:

- $\nabla_\phi G(\phi^*, a)$ is the Jacobian of $\nabla_\phi F$ with respect to $\phi$, which is the Hessian matrix of the inner objective: $\nabla^2_{\phi\phi} F(\phi^*; a, o)$.

- $\nabla_a G(\phi^*, a)$ is the Jacobian of $\nabla_\phi F$ with respect to $a$, which is the matrix of mixed partial derivatives: $\nabla^2_{a\phi} F(\phi^*; a, o)$.

Substituting these back, we get the expression for the Jacobian of the inner solution with respect to the outer parameters:

$$\frac{\partial \phi^*}{\partial a} = -\left[\nabla^2_{\phi\phi} F\right]^{-1}\left[\nabla^2_{a\phi} F\right] \tag{7}$$

This expression is computable under the proposition's assumption that the Hessian $\nabla^2_{\phi\phi} F$ is invertible at the optimum (which is guaranteed if $F$ is strongly convex in $\phi$ near $\phi^*$).

**Assembling the Final Gradient.** We can now substitute the Jacobian from Equation (7) back into the chain rule expression from Equation (6). This gives the full gradient of the term inside the expectation:

$$\frac{dF(\phi^*(a,o);a,o)}{da} = \left.\frac{\partial F}{\partial a}\right|_{\phi=\phi^*} + \left.\frac{\partial F}{\partial \phi^T}\right|_{\phi=\phi^*} \left(-\left[\nabla^2_{\phi\phi}F\right]^{-1}\left[\nabla^2_{a\phi}F\right]\right)$$

Finally, substituting this back into Equation (5) gives the complete expression for the gradient of the outer objective:

$$\nabla_a \Phi(a) = \mathbb{E}_{\epsilon \sim p(\epsilon)}\left[\left.\frac{\partial F}{\partial a}\right|_{\phi=\phi^*} - \left.\frac{\partial F}{\partial \phi^T}\right|_{\phi=\phi^*}\left[\nabla^2_{\phi\phi}F\right]^{-1}\left[\nabla^2_{a\phi}F\right]\right]$$

where all derivatives are evaluated at $\phi^*(a, g(\epsilon, a))$.

Since the proposition assumes that $p(o, x \mid a)$ and $q(x; \phi)$ are differentiable, all the required partial derivatives ($\frac{\partial F}{\partial a}$, $\frac{\partial F}{\partial \phi}$, $\nabla^2_{\phi\phi}F$, $\nabla^2_{a\phi}F$) exist and are computable. Therefore, the overall gradient $\nabla_a \Phi(a)$ is well-defined and can be estimated via Monte Carlo sampling of $\epsilon$. This confirms that the problem is differentiable and thus amenable to gradient-based optimization methods. □

### C.5 APPLICATION TO LLM ALIGNMENT

This formal template can be directly applied to the problem of LLM alignment, where the goal is to ensure an LLM's behavior conforms to a user's underlying intent.

Let the latent state be the user's true intention, $u$. The LLM cannot observe $u$ directly. Instead, it maintains a posterior belief over possible intentions, $q(u)$. The LLM's actions are clarifying questions, $\pi$, which elicit responses, $y$, from the user. The bilevel alignment process is:

$$\pi^* = \arg\min_{\pi} \mathbb{E}_{y \sim p(\cdot|\pi,u)}\left[F\left(q^*(y,\pi)\right)\right], \tag{8}$$

$$q^*(y,\pi) = \arg\min_{q} \mathbb{E}_{q(u)}\left[\ln q(u) - \ln p(y, u \mid \pi)\right]. \tag{9}$$

- The **outer loop** (Eq. 8) is the strategic decision: the LLM chooses the clarifying question $\pi^*$ that is expected to most effectively reduce its uncertainty about the user's intent $u$. By Proposition C.3, this is the question with the highest expected information gain.

- The **inner loop** (Eq. 9) is the belief update: after asking the question and receiving the user's answer $y$, the LLM updates its belief from its prior to the new posterior $q^*(u)$.

From this perspective, modern Reinforcement Learning from Human Feedback (RLHF) pipelines (Askell et al., 2021; Bai et al., 2022) can be viewed as approximate and ungrounded solutions to this same underlying objective. They are *approximate* because the reward model, trained on preference data, serves as a heuristic proxy for minimizing misunderstanding, rather than directly optimizing a formal information-theoretic objective. They are *ungrounded* because they typically do not maintain an explicit probabilistic model of user intent $p(u)$ or a formal observation model $p(y \mid u, \pi)$, which are necessary components for principled Bayesian belief updating.

## D THE INFORMATION-THEORETIC ORACLE IN DETAIL

The GTO oracle is the cornerstone of our benchmark, instantiating the perfect, unexploitable player for the OQA Information Game. Its strategy is guaranteed to minimize the expected number of questions required to identify a hidden target. This section provides a detailed breakdown of its algorithmic implementation, a formal proof of its optimality, and an analysis of its computational complexity.

### D.1 ALGORITHMIC FORMULATION AS A DYNAMIC PROGRAM

The oracle's strategy is computed using dynamic programming. The core of the algorithm is a recursive function, $C(S)$, which calculates the minimum expected cost (i.e., the number of future

---

**Algorithm 1** Optimal yes/no-query oracle (uniform prior; complexity $\mathcal{O}(d|S|^2)$)

---

    **global** table $\mathcal{C}$                                            ▷ memoized costs
1: **function** BUILDTREE($S$)                                     ▷ $S$ candidate items
2:     **if** $|S| = 1$ **then**
3:         $\mathcal{C}[S] \leftarrow 0$; **return** LEAF($S$)
4:     **end if**
5:     $bestCost \leftarrow \infty, bestNode \leftarrow$ LEAF($S$)
6:     **for all** attribute $a$ present in $S$ **do**
7:         $S^{yes} \leftarrow \{x \in S : a(x) = 1\}$;    $S^{no} \leftarrow S \setminus S^{yes}$
8:         **if** $S^{yes} = \emptyset$ **or** $S^{no} = \emptyset$ **then**
9:             **continue**
10:        **end if**
11:        $c \leftarrow 1 + \dfrac{|S^{yes}|}{|S|}C(S^{yes}) + \dfrac{|S^{no}|}{|S|}C(S^{no})$
12:        **if** $c < bestCost$ **then**
13:            $bestCost \leftarrow c$; $bestNode \leftarrow$ NODE($a$, BUILDTREE($S^{yes}$), BUILDTREE($S^{no}$))
14:        **end if**
15:     **end for**
16:     **if** $bestCost = \infty$ **then**
17:         $bestCost \leftarrow 0$
18:     **end if**
19:     $\mathcal{C}[S] \leftarrow bestCost$; **return** $bestNode$
20: **end function**
21: **function** $C(S)$
22:     **if** $S \notin \mathcal{C}$ **then**
23:         BUILDTREE($S$)
24:     **end if**
25:     **return** $\mathcal{C}[S]$
26: **end function**

---

questions) to resolve the uncertainty within a given candidate set $S$. This function is the "cost-to-go" or value function for the game state $S$.

The function adheres to the Bellman equation for this sequential decision problem:

$$C(S) = 1 + \min_{a \in \mathcal{A}(S)} \left( \frac{|S^{\text{yes}}|}{|S|}C(S^{\text{yes}}) + \frac{|S^{\text{no}}|}{|S|}C(S^{\text{no}}) \right) \tag{10}$$

Here, the '1' represents the immediate cost of asking the current question, and the minimization term represents the expected future cost under the best possible action $a$. The expectation is taken over the two possible outcomes ("yes" or "no"), weighted by their probabilities under a uniform prior. To avoid the exponential complexity of re-computing $C(S)$ for the same subsets, the algorithm uses memoization, storing the result for each unique subset $S$ after its first computation. The full pseudocode is presented in Algorithm 1.

### D.2 PROOF OF OPTIMALITY AND COMPLEXITY ANALYSIS

**Theorem D.1.** *Algorithm 1 computes the Game-Theory Optimal policy for the OQA Information Game. For a set of $N$ initial objects and $d$ attributes, its time complexity is $\mathcal{O}(d \cdot N \cdot 2^N)$ and its space complexity is $\mathcal{O}(N \cdot 2^N)$.*

*Proof.* **Optimality:** The proof of optimality rests on two pillars. First, the algorithm is a direct implementation of backward induction. The recursive function 'BuildTree' solves for the optimal policy at a given state $S$ by assuming that the policies for all subsequent, smaller states ($S^{yes}$ and $S^{no}$) have already been solved optimally. This is the essence of Bellman's principle of optimality, which is guaranteed to find the Subgame Perfect Nash Equilibrium for a finite, sequential game of perfect information. Second, as formally proven in §B the GTO policy for the OQA game is equivalent to the greedy policy of maximizing the Expected Information Gain (EIG) at each step.

The Bellman cost update in the algorithm is the mathematical dual of maximizing EIG. Therefore, the algorithm computes the GTO policy.

**Complexity Analysis:** The state space of the problem is the power set of the initial set of objects $\mathcal{X}$, which has $2^N$ possible subsets.

- **Time Complexity:** The function $C(S)$ is memoized, so its body is executed at most once for each unique subset $S \subseteq \mathcal{X}$. Inside the function, the primary work is the 'for' loop, which iterates through at most $d$ attributes. For each attribute, partitioning the set $S$ takes $\mathcal{O}(|S|)$ time. The total work is the sum of computations over all subsets: $\sum_{k=0}^{N} \binom{N}{k} \cdot d \cdot k$. This sum is equal to $d \cdot N \cdot 2^{N-1}$. Thus, the time complexity is $\mathcal{O}(d \cdot N \cdot 2^N)$.

- **Space Complexity:** The dominant factor for space is the memoization table $\mathcal{C}$, which must store a value for each possible subset of $\mathcal{X}$. The number of subsets of size $k$ is $\binom{N}{k}$. The total space required is proportional to $\sum_{k=0}^{N} \binom{N}{k} \cdot k$, which is $\mathcal{O}(N \cdot 2^N)$.

This exponential complexity makes the oracle construction intractable for very large $N$, but it is perfectly feasible for the tiers used in this paper ($N = 25, N = 100$). $\qquad\square$

# E  SUPPLEMENTAL EXPERIMENTS: SYNTHETIC DATASETS

To provide a more rigorous "stress test" of the models' underlying strategic reasoning, we designed two synthetic datasets that ablate all real-world semantic information. This isolates pure planning ability from heuristic pattern matching based on linguistic priors learned during pretraining.

.

## E.1  METHODOLOGY

To probe how frontier LLMs behave when no natural-language priors are available, we generated two purely synthetic guessing corpora of size 25 and size 100. Each object is identified only by a hexadecimal key and a 10-dimensional Boolean attribute vector. The generator below enumerates all $2^{10}$ attribute combinations, then samples a subset:

```python
import itertools, random

def generate_synthetic_data():
    attributes = ["a", "b", "c", "e", "f", "g", "h", "i", "j", "k"]
    synthetic_data = {}
    for i, combo in enumerate(itertools.product([False, True],
                                                repeat=len(attributes))):
        synthetic_data[f"{i:10x}"] = dict(zip(attributes, combo))
    return synthetic_data

data = generate_synthetic_data()

subset_25  = random.sample(list(data.keys()), 25)
subset_100 = random.sample(list(data.keys()), 100)
```

Apart from object names, the evaluation protocol is identical to Section 4. Attribute vectors remain unique, so the oracle achieves the same theoretical optimum as in the realistic domains.

Without memorable names, models often drop viable candidates from their implicit belief state; a single reminder prompt usually suffices for the 25-object set, whereas two or more prompts are needed once the pool grows to 100. Across all seven LLMs the mean query counts and entropy trajectories stay above the oracle's, and the synthetic curves exhibit the bumps seen in Figures 4a and 4b; this gap amounts to roughly one to three extra questions on average compared with realistic datasets. Finally, the absence of linguistic cues magnifies familiar shortcomings on algorithmic subtasks such as counting and set manipulation, further undermining query optimality.

Synthetic datasets matter because they can be scaled programmatically to thousands of objects and attributes, synthetic worlds offer an open-ended testbed for measuring pure information-gathering ability. Closing the synthetic–oracle gap therefore remains an attractive target for future LLM research.

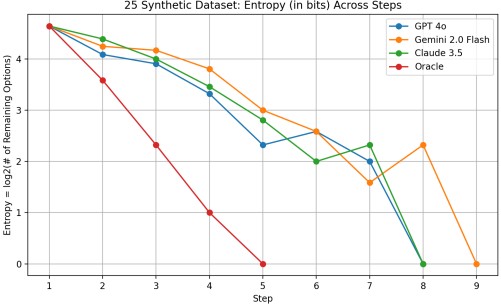 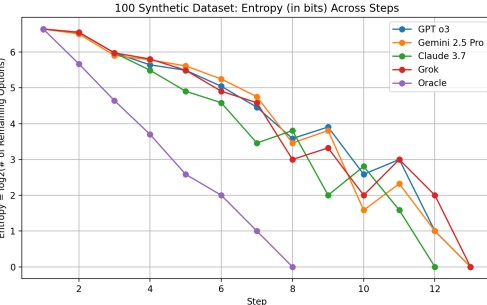

(a) Posterior entropy on the 25-object synthetic set     (b) Posterior entropy on the 100-object synthetic set

Figure 4: Entropy versus dialog turn on synthetic datasets. Each curve shows the integer floor of the mean over five random targets; characteristic bumps appear when missing candidates are rediscovered after reminder prompts. The oracle curve marks the information-theoretic optimum.

