# OpenReview forum: "The Information Game: Active Inference as Bilevel Optimization and a Game-Theoretic Benchmark for LLM Inquiry"
_ICLR.cc/2026/Conference — ICLR 2026 Conference Withdrawn Submission_

### Official Review · Reviewer_53vQ · 2025-10-26

**Soundness:** 2
**Presentation:** 2
**Contribution:** 2
**Rating:** 2
**Confidence:** 3

**Summary:**

This paper proposes a novel task for LLM, which is optimal querying for information gathering and elimination of uncertainty. Compared to prior optimal querying tasks such as twenty questions, this task is based on game theory and has better theoretical, quantitative properties. The paper gives theoretical proof on such type of task, proposes a query efficiency benchmark as an instance of the task, and conducts empirical study on this benchmark with several state-of-the-art models, which shows state-of-the-art models are suboptimal.

**Strengths:**

1. The paper provides a detailed description on the experiment, and the authors also provide their code, experiment logs and dataset. The reproducibility of their work seems promising.

2. The general motivation and idea of this paper is clearly conveyed: to quantitatively measure LLM's ability to curate queries for gathering information.

3. The related work seems well-rounded, with investigations on many related areas to this work.

**Weaknesses:**

1. I am not sure about the necessity of involving game theory in this proposed task. The paper models the task as a two-player zero-sum game, with the inquirer asking question and the responder answering question. However, the responder does not seem to have any alternative strategy rather than answering "yes" or "no" honestly according to fact (as mentioned in line 215). In this case, the game degrades into a single-agent task where the responder is simply part of the static environment, and proposition B.15 can be also modeled as finding the shortest path from the full set to the node with the subset that only contains the nature's choice in definition B. 13 (ii), which can be done by dynamic programming (backward induction of Appendix B.5). This seems to be different from the AKQ game mentioned in the paper, where the two agents both have degree of freedom on their actions.

2. If the selection of nature (definition B.13) and attributes are uniformly randomly sampled (line 190), then I would assume randomness plays an important role in the performance of the models as the difficulty can significantly vary with the attribute/answer chosen. However, there is no error bars, standard deviations, seeds or confidence interval reported in the main results.

3. The ability that the benchmark is trying to test is intertwined. The author mentions in line 1453 that "Without memorable names, models often drop viable candidates from their implicit belief state". However, the purpose of synthetic dataset is only to remove linguistic priors (line 1426), not to test the model's long-context memorization ability. Thus, when there exists an optimality gap, it is hard to say which factor is the bottleneck - this obfuscates the insight that we can gain from the benchmark and how we can build a better model to address the task. The paper also does not try to address this issue by ablation (which should be in the main paper as currently it has one unused page).

4. The empirical analysis does not provide much insight on "how we can build a better model" (with either successful or failed attempts) beyond "state-of-the-art models are suboptimal" (model dropping viable candidates is one, but this alone is insufficient). The message "current state-of-the-art cannot solve" can often be a short-lived message - cutting-edge LLMs improves very quickly on super-hard benchmarks such as Humanity's Last Exam [1] and ARC-AGI-2 [2]. The behavior analysis and attempted solutions, however, are much more valuable. Besides, the models does not seem to be too far away from optimal - is "a planning gap of 1-2 queries" unacceptable?

**Minor Weakness**

In line 177, 190 and line 260, periods are missing.

**References**

[1] L. Phan et al. Humanity's Last Exam. ArXiv: 2501.14249, 2025.

[2] https://arcprize.org/arc-agi/2/

**Questions:**

I have one question: Did the authors find any behavioral difference between the synthetic dataset and the dataset with real-life semantic priors (maybe besides that the synthetic items are easier to be forgotten as suggested in line 1452 - for this part, is there any quantitative metric?)

---

### Official Review · Reviewer_JAUX · 2025-10-31

**Soundness:** 2
**Presentation:** 2
**Contribution:** 2
**Rating:** 2
**Confidence:** 3

**Summary:**

This paper introduces the Optimal Question Asking (OQA) benchmark, which measures how efficiently LLMs reduce uncertainty through yes/no questions about hidden objects. The authors propose to view the OQA game objective as a game-theory optimal solution minimizing the number of queries. Based on this, the oracle is a dynamic programming-based solution that makes the most informative query at every step, to minimize the number of queries. They authors evaluate several LLMs and measure the gap between the number of queries used and the optimal number of queries, showing that LLMs show suboptimal query usage.

**Strengths:**

- The paper studies an important problem: Benchmarks measuring query efficiency in information gathering are an under-explored area of research, and as the authors argue, relevant to many real world scenarios such as alignment, tutoring and clarification seeking.
- The authors define an information-theoretic lower-bound that is conceptually useful
- Experiments cover real world data and sythetic data

**Weaknesses:**

- I'm not sure that the optimal solution as found by dynamic programming is actually feasible to be computed by humans or LLMs through reasoning, without access to computational tools. This makes it difficult to interpret what exactly the measured planning gap actually implies.
- The evaluation is limited to object-guessing games, which is a narrow subset of information gathering tasks.
- The connection between GTO, EIG maximization, and EFE minimization is difficult to follow, and overall I could not understand the purpose of section 3's bilevel optimization view, as it does not seem to effect any of the following sections or the paper's experiments.
- The experiments use the chat UI rather than APIs, without any details such as temperature, prompts, or usage of reasoning capability, making the experiments difficult to reproduce.
- The organization of the paper is difficult to follow, with many results and analysis pushed to the appendix without proper integration in the main text (e.g. experiments on synthetic data)

**Questions:**

- Did the authors consider establishing human baselines, or LLM baselines with computational tools?
- Does the "implicit belief state" refer to computations internal to the LLM, or does it refer to some kind of intermediate text such as the model's thought or reasoning outputs?
- What are some possible ways to extend the framework to more domains not restricted to binary choices?

---

### Official Review · Reviewer_Lf2E · 2025-10-31

**Soundness:** 1
**Presentation:** 1
**Contribution:** 1
**Rating:** 2
**Confidence:** 4

**Summary:**

- This paper proposes a new benchmark for measuring an LLM agent’s information-seeking strategy. The benchmark consists of 25 or 100 objects that can be queried through a series of yes/no questions and answers. The simplicity of the domain and the binary question-answering setup allow for comparison with an oracle strategy that maximizes expected information gain at each step.
- The benchmark uses the mean number of queries (fewer indicates better “active inference”) to measure the effectiveness of different existing LLMs.
- The paper formulates active inference as a bi-level optimization problem involving belief updating and decision-making to minimize variational free energy.

**Strengths:**

While this paper discusses an interesting problem of active information gathering in multi-turn settings( where both the agent’s information-gathering and response-generation abilities are crucial to success) there is a significant gap between Sections 2–3 and the empirical results in Section 4, which limits the paper’s contribution and originality. Please see Weaknesses section for further details.

**Weaknesses:**

Disconnect between framework and theory (sections 2 & 3) and empirical demonstrations (section 4):
- Sections 2 and 3 focus on formalizing the bi-level optimization and propose, in Proposition 3.2, a structure that can be used for gradient-based training. However, the empirical results in Section 4 only evaluate the zero-shot capabilities of existing models using the prompt in Fig 1. It is unclear how the experiments connect to the proposed bi-level optimization framework beyond the comparison with the oracle strategy, as the oracle selects according to the maximal expected information gain (which would be optimal under the bi-level framework). The training objective (from Proposition 3.2) is neither implemented nor referred to in the experiments.
- The contributions of this paper could be significantly strengthened if the proposed bi-level approach were either (a) used to improve models via fine-tuning or (b) incorporated into the models’ inference-time action selection. This would help establish the work's originality and significance.

Lack of empirical evidence and statistical significance:
- Table 2 is generated from "the mean over five random targets per tier" but no standard deviation across the five targets is shown. Also testing with only five targets is too small to determine whether the reported results are statistically significant or meaningful.
- Given this small sample size, the gap of 1-2 queries between the tested models and the oracle strategy is too small to make the empirical claims (e.g., "Lighter models close the gap to within two questions on the 25-object tier ... larger models ask more optimal questions on average and maintain a near-optimal belief state even late in the dialogue" in Lines 282-285) convincing.

Clarity:
- Terms in proposition 3.2 can be formally defined.
- Consistent notations can be used through Section 3: for example, $H(P(x))$ in line 161, if referring to states, could use $S$ instead of $X$ for consistency.
- Section 3 would benefit from a formal problem setup section and introduction of relevant notations prior to mentioning any propositions for readability.

Potential consideration for adding real-world motivated complexities to the tasks in addition to the object guessing games:
- While the motivation for keeping the benchmark tasks simple and tractable is clear, the problems can be made more complex by adapting existing real-world datasets or multi-turn datasets from education (e.g., Shani et al., 2024; Wan et al., 2025). The actions can still be constrained to binary options (querying for yes or no answers) if tractability is a concern.

Shani et al., 2024. Multi-turn Reinforcement Learning from Preference Human Feedback. https://arxiv.org/abs/2405.14655

Wan et al., 2025. Enhancing Personalized Multi-Turn Dialogue with Curiosity Reward. https://arxiv.org/html/2504.03206v2

**Questions:**

- A stronger connection could be made between Sections 2 & 3 (formulation of the bi-level optimization problem for active inference) and Section 4 (experiments). Please see the first point under "Weaknesses."
- The empirical results could be strengthened by evaluating larger samples and reporting statistical significance and standard deviation in addition to the mean. This is especially important given the small gap between the tested models and the oracle (only 1-2 queries).

---

### Official Review · Reviewer_nWPQ · 2025-11-05

**Soundness:** 1
**Presentation:** 2
**Contribution:** 3
**Rating:** 2
**Confidence:** 2

**Summary:**

The paper frames optimal question asking as a sequential information game. Beliefs update via variational inference (inner loop). Question selection minimizes Expected Free Energy (EFE) is the outer loop.  Under their assumptions, this reduces to maximizing Expected Information Gain (EIG).

The paper introduce OQA, a small controlled benchmark where an agent identifies a hidden target (25 or 100 objects) by asking yes/no attribute questions on a list of attributes. They provide an oracle that greedily maximizes EIG each turn which is “game-theory-optimal (GTO)” and measure LLMs’ “planning gap” to this oracle on real (Animals, Cars, Places) datasets along with synthetic datasets. Empirically, frontier models trail the oracle by about 1–3 questions on 25 and 100-object tier tasks across benchmarks.

**Strengths:**

**Clear, focused benchmark for inquiry.** OQA isolates question selection rather than answer quality. The instruction template and dataset design are simple and well motivated (Sec. 4).

**Clear lower bound and metric.** The paper provides an oracle and a crisp metric (gap to the oracle in questions), making results easy to interpret. Figures 2–4 make the effect visible: models reduce entropy like binary search early, then loses efficiency later.

**Weaknesses:**

**[Critical] Cost depends on structure, not only set size/entropy.** Step 1 (“Cost depends only on |S|”) does not hold when the question set is restricted to a fixed attribute table (the benchmark’s setting). Two subsets with the same size can admit different partitions and thus different optimal costs. Therefore, C(S) is not a function of ∣S∣ alone. How does the theory translate the OQA benchmark?

**[Critical] “MDP with nature” formalism.** The paper presents a two-player zero-sum game with SPNE terminology (§2, App. B), yet isn't the “Responder” is a deterministic channel with no strategy? The current framing seems misleading/over-theorizing.

**[Critical] Greedy EIG can be suboptimal for expected depth.** With restricted tests, locally minimizing expected posterior entropy need not minimize the expected number of questions. Classical decision-tree results and counterexamples show that greedy information gain can yield deeper trees in expectation [1], [2].

- [1] Constructing Optimal Binary Decision Trees is NP-Complete
- [2] Performance Bounds on the Splitting Algorithm for Binary Testing

**[Critical] Conflicting complexity claims.** Section 6 and Algorithm 1 state the oracle DP runs in O(d|S|^2) and is tractable up to ~100 items. Appendix D (Thm. D.1) gives O(d\cdot N \cdot 2^N) time and O(N\cdot 2^N) space -- which seems to be standard for memoizing all subsets. These statements plainly conflict. Could the authors reconcile these two aspects?

**[Major] Unclear experimental procedure.** Algorithm 1 does not make clear how you select the maximal-information-gain question in the benchmark’s restricted attribute set. Could you describe how the oracle was computed and how the attributes were queried? Generally, experimental details are lacking -- It is unclear how scores in the table are produced.
eg.
- The algorithm seem to require each model to output a probability distribution over 100 objects after every turn; please detail the prompting, decoding, calibration strategies. (2)
- The paper states “We can stop when there are no more distinguishing attributes” but ANIMALS contains duplicates (p. 4). How does the paper count queries when equivalence classes remain and confirm consistent handling across models?

**[Major] Resolution for proposed procedures.** The paper proposes procedures to improve information seeking. Do the experiments have enough resolution to detect the claimed improvements? What are the confidence intervals across the models tested?

**Questions:**

Please address the weaknesses.

---

### Note · Authors · 2025-11-15

I have read and agree with the venue's withdrawal policy on behalf of myself and my co-authors.